# Physicochemical Composition and Fatty Acid Profile of Goat Kids’ Meat Fed Ground-Corn-Grain Silage Rehydrated with Different Additives

**DOI:** 10.3390/ani13010031

**Published:** 2022-12-21

**Authors:** Luciana V. Diogénes, Ricardo L. Edvan, Elisama dos S. Medeiros, José M. Pereira Filho, Juliana P. F. de Oliveira, Edson C. Silva Filho, Layse M. G. Ramos, Kevily H. de O. S. de Lucena, Marcos J. Araújo, Ronaldo L. Oliveira, Elzania S. Pereira, Leilson R. Bezerra

**Affiliations:** 1Animal Science and Health Graduate Program, Federal University of Campina Grande, Patos 58708-110, Brazil; 2Animal Science Department, Federal University of Piaui, Teresina 64049-550, Brazil; 3Animal Science Department, Federal University of Sergipe, Nossa Senhora da Glória 49680-000, Brazil; 4Interdiscisciplinary Laboratory of Advanced Materials, Chemistry Department, Federal University of Piaui, Teresina 64049-550, Brazil; 5Animal Science Department, Federal University of Bahia, Salvador 40170-110, Brazil; 6Animal Science Department, Federal University of Ceara, Fortaleza 60020-181, Brazil

**Keywords:** cactus pear mucilage, fatty acids, goats, shear force, whey

## Abstract

**Simple Summary:**

Corn is one of the most produced grains in the world, being the primary energy concentrate used in animal feeding due to its nutrition quality and high cultivation potential, which make it the main source of starch in the diets of ruminant animals. We compared three different moisture additives to conserve rehydrated corn-grain silage, namely water, cactus mucilage, and whey, and replaced ground corn (GC) with rehydrated corn-grain silage in the finishing of kids’ diet. Cactus pear cladode mucilage was a more efficient moisture additive than water and whey for ground-corn silage. Thus, it is recommended to replace ground corn with rehydrated corn-grain silage with mucilage (RCSmucilage) in 415 g/kg because it provides better animal performance than ground corn (GC)—along with having a similar intake and digestion to it—due to better conservation of the silage at a lower cost when compared to commercial additives.

**Abstract:**

The effects of the replacement of dry ground corn (GC) with corn-grain silage rehydrated with water (RCSwater), cactus pear mucilage (RCSmucilage), and whey (RCSwhey) on the growth, physicochemical composition, and fatty acid profile of goat kids’ meat were investigated. Thirty-two crossbred goat kids (16.4 ± 2.50 kg initial weight) were assigned in a randomized block design with four treatments and eight repetitions. The NDF intake of goat kids fed with RCSmucilage was higher in comparison to RCSwater and RCSwhey (*p* = 0.0009). The dietary replacement of GC by RCSmucilage increased the final weight (*p* = 0.033) and meat-cooking losses (*p* = 0.0001) of kids. The concentrations of oleic (*p* = 0.046), 11,14-eicosadienoic (*p* = 0.033), and EPA (*p* = 0.010) were higher in the meat of kids fed with RCSmucilage and RCSwhey, and the α-linolenic concentration was higher (*p* = 0.019) for animals feeding with RCSmucilage. Meat from kids fed with RCSwhey presented the lowest ∑SFA and the highest ∑MUFA. In contrast, the ∑PUFA (*p* < 0.012) was higher for goats fed with RCSwater. The ∑ω3 (*p* < 0.0001) was higher in animals fed with RCSmucilage and RCSwhey. Desirable fatty acids were higher (*p* = 0.044) in animals fed with RCSmucilage and RCSwhey, and the atherogenicity (*p* = 0.044) and thrombogenicity (*p* < 0.0001) indexes were lower for goats fed the RCSwhey diet. The enzymatic activities of Δ^9^desaturase (C16) were higher (*p* = 0.027) in goat kids fed with GC and RCSmucilage, and Δ^9^desaturase (C18) was higher (*p* = 0.0497) when goats were fed with RCSmucilage and RCSwhey. Elongase activities were higher (*p* = 0.045) in goat kids fed with GC and RCSwater. The total replacement of GC by RCSmucilage is recommended in the diet of goat kids due to improvements in the weight gain and proportion of desirable fatty acids in the meat. In addition, RCSmucilage promoted better conservation of the silage at a lower cost when compared to commercial additives.

## 1. Introduction

The consumption of goat meat is currently increasing worldwide, and the quality of meat is a decisive parameter, as it determines the interest and acceptability of the product, since the meat has a low content of saturated fatty acids and cholesterol and high concentration of fatty acids, and these are attributes that have been increasingly requested by the consumers [1]. Goat performance is the basis for meat production, whereas the amount and site of fat in the carcass influence its quality. In this sense, the adoption of confinement systems for goats in times of low feed production can overcome that problem, thus optimizing meat production throughout the year, noting that investments are needed, especially in the diet [2].

Corn is one of the main grains used in animal and human diets and has a fundamental role in the composition of ruminants’ feed as a source of energy, being widely used in feedlots due to its high availability in the market and nutritional value [3]. To improve its use, corn can be processed, increasing its energy availability for ruminants. There are several ways of processing these grains that can result in a greater availability of starch and thus promote better use of this cereal, mainly milling, silage, and rehydration of corn grains [4,5].

The rehydration of corn-grain silages to achieve adequate moisture levels to improve fermentation profile is a good strategy to improve the utilization of corn grain by the ruminants. This technique can improve starch digestibility and reduce the integrity of the protein matrix, thus providing greater access of microorganisms and intestinal enzymes to the starch granules [6,7]. Consequently, glucose uptake is increased indirectly through gluconeogenesis, which uses propionate as the primary substrate, or directly through glucose uptake in the small intestine [8,9]. Thus, ground-corn supplementation provides added digestible energy and protein and can help satisfy dietary requirements to improve nutrient-use efficiency for maximal performance by grazing livestock [10].

In the rehydration process of corn grain, water is used as a moisture additive prior to the ensiling process. However, other moisture additives, such as whey and cactus pear mucilage, have been studied [3,11]. These additives are used to provide moisture to corn grain, especially in regions where water availability is restricted, also acting as a source of soluble carbohydrates, besides avoiding the disposal of the whey in the environment [12], thus establishing a better fermentation of the ensiled material. Information on the use of different moisture additives during corn-grain ensiling and studies on the effect of rehydrated corn-grain silage on goat-meat quality are scarce in the literature. Thus, it was hypothesized that ground-corn-grain silage rehydrated with natural additives, especially cactus pear mucilage, could replace ground corn in the concentrate, maximizing the intake due to the greater availability of the starch and, consequently, the production and meat quality.

Thus, this study was conducted to evaluate the effects of water, cactus pear mucilage, and whey as wet additives to corn-grain silage on physicochemical composition and fatty acid composition of goat meat.

## 2. Materials and Methods

The use of animals in the experiment was approved by the Animal Use Ethics Committee of the Federal University of Campina Grande, Paraiba, Brazil (Protocol number 07/2019), following the guidelines of the National Council of Control of Animal Experimentation (Brasília, Brazil; CONCEA).

### 2.1. Animals, Treatments and Experimental Diets

Thirty-two crossbred 6-months-old non-castrated male goat kids with an average weight of 16.4 ± 2.5 kg were distributed in a randomized block design, with the initial weight of the animals being the criterion for the formation of the blocks, with four treatments and eight replications. All animals were initially vaccinated and dewormed and housed individually in stalls with drinking and feeding troughs. 

The treatments consisted of a control, which had ground corn (GC); corn-grain silage rehydrated with water (RCSwater); corn-grain silage rehydrated with cactus pear mucilage (RCSmucilage); and corn-grain silage rehydrated with whey (RCSwhey). 

The experimental diets had a 40:60 ratio of roughage: concentrate in the form of total mixture (Table 1) and were formulated to meet the weight-gain requirements according to NRC [13], aiming for an average daily weight gain of 200 g/day. The rehydrated ground-corn-grain silage replaced the ground corn grain in the concentrate and was incorporated only at the moment of feeding. The roughage consisted of chopped Tifton-85 hay. The other ingredients that made up the concentrate were soybean meal, ground corn grain, mineral mixture, and urea (Table 1). The experiment lasted 71 days, including 14 days for the animals to acclimatize to the environment, handling, and diet. The diets were offered twice daily, at 8 a.m. and 3 p.m. Intake was measured daily, maintaining a surplus of 10%. Water was provided at will.

### 2.2. Silage Making and Analyses

Three moisture additives were used in the corn grain (water, cactus pear cladode mucilage, and whey) in the proportion of 750 g/kg of dry grain to 250 g/kg of moisture additive, reaching a DM content around 650 g/kg in the rehydrated silage.

The corn was ground in a forage chopping machine (TRF 700G Trapp^®^, Santa Catarina, Brazil) coupled to a sieve with holes of 5.0 mm. The water used was treated and of good quality. The whey was used in natura immediately after enzymatic coagulation of cow’s milk, derived from regional artisan cheese making. The whey from a single day’s production was used to make the silage. 

The cactus pear cladodes (*Opuntia stricta* (Haw.) Haw (Cactaceae) were harvested at the joints after one year of regrowth, selected, and cut with a 40 cm blade (Tramontina^®,^ Campina Grande, Brazil). After cutting, the material was spread in a covered shed and ground in a forage chopping machine (TRF 700G Trapp^®^, Santa Catarina, Brazil), resulting in a mucilage-like mass of pasty consistency. 

After grinding, the ground corn grain was rehydrated with different additives moisture additives (on as-fed basis), which were subsequently homogenized. The resulting mass was ensiled in experimental silos, which were buckets (10 replications each) with a height of 0.20 m and diameter of 0.17 m (4 L capacity). One kilogram of absorbent material (coarse sand) was placed at the bottom of each silo to absorb possible effluents from the fermentation process. Above the sand layer, a non-woven synthetic fiber was used to separate the silage from the sand.

The compaction of the ground-corn-grain silage was performed in layers approximately 5 cm thick, until a density of approximately 850 kg/m^3^ was obtained. The silos were closed with plastic lids that contained Bunsen-type valves to exhaust the gases produced during the fermentation process, and then they were opened after 230 days.

Samples were collected, and gas loss (GL) and effluent loss (EL) were assessed. The dry-matter recovery (DMR) was obtained according to the equations provided by Reference [15]. The pH and ammonia nitrogen concentration (N-NH_3_) of the silages were determined according to the methodology described by Reference [16].

In the same proportions as the experimental silos, the ground-corn-grain silage with the different moisture additives was placed in plastic barrels (Polyembalagens^®^, São Paulo, Brazil) with a capacity of 240 L. The material was compacted with a manual compactor and closed with airtight lids. The barrels were then stored in the shade, at a temperature between 20 and 25 °C, and opened after 230 days to be used as part of the animals’ diet.

### 2.3. Determination of Chemical Composition

Samples of the diet ingredients and leftovers were dried in a forced ventilation oven (55 °C for 72 h) and then ground in a Willey-type knife mill into 1 mm particles.

The chemical composition followed the recommendations of Reference [17] for dry matter (method 934.01), ash (method 942.05), crude protein (method 968.06), and ether extract (method 920.39) contents. For the determination of neutral detergent fiber (NDF) and acid detergent fiber (ADF), the methodologies described by Van Soest et al. [18] were used. For the analysis of the NDF, three drops (50 μL) of α-amylase were added per sample in the washing with detergent, as well as in the water. The NDF content was corrected for ash and protein (NDF_ap_) by following the methodology described by Licitra et al. [19] wherein the neutral detergent residue was burnt in a muffle furnace at 600 °C for 4 h, and the correction for the protein was carried by not considering the neutral detergent insoluble protein. The acid detergent lignin (ADL) was obtained by following the methodology described by AOAC (method 973.18) [17].

Non-fiber carbohydrates (NFCs) were calculated by considering in the calculation the NFC value corrected for ash and protein [20]. The metabolizable energy (ME) was obtained from the total digestible nutrients (TDNs) of the diets, considering that 1 kg of TDN corresponds to 4.409 Mcal of digestible energy (ED), which, in turn, presents 82% of use efficiency by the animal and represents the ME of the feed [21] according to Reference [14].

### 2.4. Slaughter and Sample Collection of the Muscles Longissimus Lumborum

At the end of the experiment, all goat kids were weighed before slaughter (final body weight) after a 16-h fast. The animals were stunned by electronarcosis (minimum current of 1.25 amperes), and then bleeding was performed by sectioning the jugular veins and carotid arteries, followed by skinning and evisceration. The carcasses were stored in a cold chamber (4 °C) for 24 h and then hung by hooks through the calcaneal tendon. 

The pH was measured 24 h after carcass cooling at the longissimus lumborum (LL) between the 12th and 13th ribs on the left half of the carcass of each animal. The pH of the meat was measured by using an insertion pH meter (Testo Instrument Co., Ltd., Lenzkirch, Germany). Then the left and right LL muscles were dissected and stored in a freezer (−20 °C) for backup and FA profiling.

### 2.5. Physicochemical Properties of the Muscle Longissimus Lumborum

Meat-color determination was performed immediately after the measuring of the pH in a cross-section in the LL muscle, at temperatures between 6 and 7 °C [22], for 40 min before readings to expose myoglobin to oxygen [23]. The color of samples was determined by using a colorimeter (Chroma meter, CR-410, Tokyo, Japan) equipped with a standard D65 light source, using a standard observer’s 2° position with a pulsed xenon lamp and an 8 mm reading surface area. The lightness (L*), redness (a*), and yellowness (b*) for each sample were recorded in triplicate for each sample, and the average values were recorded. The saturation index (chroma, C*) was determined from the a* and b* data, according to the formula C* = [(a*)^2^ + (b*)^2^]^0.5^, according to Reference [24].

The water-holding capacity (WHC) of LL muscle was determined by using meat samples of approximately 300 mg, which were placed between circular filter papers (Albert 238, 12.5 cm in diameter, São Paulo, Brazil) and loaded with 3.6 kg for 5 min [25]. The WHC was obtained by the difference in weight of the samples before and after the application of the load, with values expressed as g/100 g.

For cooking weight loss (CWL%), the samples were cooked individually in plastic bags immersed in a water bath at 75 °C until they reached an internal temperature of 71 °C, which they were then held at for 20 min. During cooking, the core temperature of the samples was tracked by using a digital thermometer (DM6801A, Shenzhen Victor Hi-tech Co., Ltd., Shenzhen, China). Then cooking loss was calculated as the g/100 g of exudate before and after heating [26].

In the following procedure, goat-kid meat samples were brought to room temperature prior to Warner–Bratzler Shear Force (WBSF) analysis. About 6 to 8 cylindrical cores that were 1.27 cm in diameter and were parallel to the muscle fiber orientation were removed from each muscle sample [26]. A single peak-shear-force measurement was obtained for each core, using a Warner-Bratzler meat shear machine (Texture Analyzer TX-TX2, Mecmesin, NV, USA), and an average shear force was calculated and recorded for each muscle sample. The results were expressed as load in Newton (*N*) [27].

### 2.6. Fatty Acids Profile of Longissimus Lumborum

The samples to determine chemical composition and the fatty acid (FA) profile of LL muscle were freeze-dried for 72 h; ground; and analyzed for moisture content (method 967.03), protein (method 981.10), lipid (method ISO 1443.1973), and ash (method 942.05) in accordance with the recommendations of Reference [17].

Lipids were extracted by using a chloroform–methanol mixture [28], and fatty acid methyl esters (FAMEs) were extracted by following the ISO 5509 method (ISO, 1978), using n-hexane, methanol, and KOH. The total chromatography was divided into 4 heating cycles as follows: 100 °C (5 min), 190 °C (5 °C/min for 5 min), 220 °C (2 °C/min for 5 min), and 240 °C (5 °C/min for 5 min). Helium gas was used as the carrier gas at a flow rate of 1.0 mL/min, split 1:20, and the temperature of both the injector and detector was 260 °C. Fatty acid identification was based on the retention time of the methyl esters of fatty acid standard (ethyl palmitate).

The quantification and determination of fatty acids in the different diets (Table 2) and the meat from goat kids were performed in triplicate, using a gas chromatograph–mass spectrometer (GCMS-QP2010 SE), with an RT-x Wax Polietileno Glicol column 30 m length with a 0.25 mm in diameter and a film thickness of 0.25 μm (Supelco SP-24136, Sigma-Aldrich, São Paulo, Brazil). 

The total chromatography was divided into 4 heating cycles as follows: 100 °C (5 min), 190 °C (5 °C/min for 5 min), 220 °C (2 °C/min for 5 min), and 240 °C (5 °C/min for 5 min). Helium gas was used as the carrier gas at a flow rate of 1.0 mL/min, split 1:20, and the temperature of the injector and detector was 260 °C. Fatty acid identification was based on the retention time of the methyl esters of fatty acid mixed standard (C4-C24, Sigma-Aldrich, São Paulo, Brazil).

The results were quantified by normalizing the areas of the methyl esters and are expressed as g/100 g fatty acids methyl esters (FAMEs). The totals for saturated fatty acids (SFAs), monounsaturated fatty acids (MUFAs), and polyunsaturated fatty acid (PUFAs), as well as the MUFA:SFA, PUFA:SFA, PUFA:MUFA, and *n–6:n–3* ratios, were calculated from the identified fatty acid profiles. The nutritional quality of the lipid fraction of the goat meat was evaluated from the atherogenicity index (AI) equation AI = [(C12:0 + (4 × C14:0) + C16:0)]/(ΣMUFA + Σn–6 + Σn–3) [29], and hypocholesterolemic and hypercholesterolemic ratio (h:H index) by equation = (C18:1 cis–9 + C18:2 n–6)/(C14:0 + 16:0) [30]. The desirable fatty acids (DFAs) were evaluated according to the equation of Rhee [31].

The activities of Δ^9^-desaturase C16 (D9C16), da Δ^9^-desaturase C18 (D9C18), and elongase were estimated according to Smet et al. [32] from the following equations: D9C16 = [C16:1/(C16:0 + C16:1)] × 100, D9C18 = [(C18:1 cis–9)/(C18:0 + C18:1 cis–9)] × 100 and elongase = [(C18:0+ C18:1 cis–9)/(C16:0 + C16:1 + C18:0 + C18:1 cis–9)] × 100.

### 2.7. Statistical Analysis 

A randomized block design with four treatments and eight replications per treatment was used to evaluate meat and silage quality. 

The data were subjected to analysis of variance, and the treatment means were compared thorough the Tukey’s test at 5% probability, using PROC MIXED of the Statistical Analysis System software, SAS^®^, version 9.4 [33]. The model used includes the random effect of blocks and the treatments as a fixed effect.
Yijk = μ + τi + βj + τβij + εijk,(1)
where Yijk = value k observed in the experimental unit that received treatment i, replication j; μ = general mean common to all observations; τi = effect of treatment i; βj = effect of block j; τβij = effect of interaction between treatment i and block j; and εijk = random error with mean 0 and variance σ².

## 3. Results

### 3.1. Growth and Physicochemical Composition of Goat Meat 

The dry matter (DM) intake was higher for goat kids fed with RCSmucilage (*p* = 0.035), and the other treatments did not differ among them (Table 3). The NDF intake for goat kids fed with RCSmucilage was higher when compared to RCSwater and RCSwhey (*p* = 0.0009). Replacing GC with RCSmucilage in the goat kids’ diets promoted a higher final body weight (*p* = 0.033). The intake of crude protein and the metabolizable energy were similar among goat kids fed the different diets.

There was no significant difference (*p* > 0.05) in the 24 h postmortem pH of the meat of goat kids fed with diets (Table 4). The data revealed that the color indices L*, a*, b*, and chroma were similar among the goats fed with the different diets. There was no effect between treatments for water-holding capacity and shear force (*p* > 0.05). The cooking weight loss was higher for goat kids fed with RCSmucilage and GC (*p* = 0.0001) when compared to the other rehydrated silages. Regarding the chemical composition, there was no effect (*p* > 0.05) on the moisture, crude protein, and lipids of the meat caused by the different diets. However, the ash content was higher (*p* = 0.027) in the meat of animals fed with RCSwater and RCSmucilage.

### 3.2. Fatty Acids Profile of Goat Meat 

The fatty acid (FA) profile of the intramuscular fat from the LL muscle of the goat kids is presented in Table 5. The concentrations of the saturated fatty acids (SFAs), butyric (C4:0), caproic (C6:0), caprylic (C8:0), myristic (C14:0), and pentadecanoic (C15:0) were influenced by the different diets. Ground corn and RCSwater diets increased the concentration of C4:0 (*p* = 0.018). In contrast, C6:0 was higher in the meat of animals fed with ground corn and RCSmucilage (*p* = 0.045). The RCSmucilage and RCSwhey diets reduced the saturated fatty acids contents of C8:0 and C14:0, respectively. The concentration of C15:0 was higher (*p* = 0.029) in the meat of goat kids fed with ground corn. However, feeding dried or ensiled corn did not influence the proportion of two major SFAs, namely palmitic acid (C16:0) and stearic acid (C18:0).

The concentration of the monounsaturated fatty acid (MUFA) oleic (C18:1 ω9) (*p* = 0.046) was higher (*p* = 0.081) in the meat of goat kids fed with RCSmucilage and RCSwhey, whereas erucic (C22:1 ω9) (*p* = 0.0001) only in the meat of goat kids fed with RCSwhey. 

Regarding the polyunsaturated fatty acids (PUFAs), the concentrations of α-linolenic (C18:3 ω3) was higher (*p* = 0.019) in the meat of goat kids fed with RCSmucilage, while 11,14-eicosadienoic (C20:2 ω6) and EPA (C20:5 ω3) where higher with RCSmucilage and RCSwhey diets, and di-homo-α-linolenic (C20:3 ω3) only with the RCSwhey diet.

The intramuscular fat of the muscle LL (*p* = 0.012) of goat kids in the RCSwhey dietary treatment obtained the lowest sum of SFA, and the highest sum of MUFA. The sum of PUFA, in contrast, (*p* < 0.0001) was higher in the meat of the goat kids fed with RCSwater. The ω3, (*p* < 0.0001) was also affected by the diets, which was higher in the animals on the RCSmucilage and RCSwhey dietary treatments. 

The MUFA:SFA ratio in the LL muscle (*p* = 0.017) of the goat kids was lower in the RCSwhey treatment and the PUFA:SFA ratio (*p* < 0.0001) in the RCSwater and RCSwhey. The ω6:ω3 ratio was higher (*p* = 0.006) in the meat of animals fed with GC and RCSwater. 

Nutraceutical compounds, desirable fatty acids (DFAs), the atherogenicity index (AI), and the thrombogenicity index (TI) were also affected by the diets. DFAs were higher (*p* = 0.044) in the meat of goat kids fed with RCSmucilage and RCSwhey. The AI (*p* = 0.044) and TI (*p* < 0.0001) were lower for goat kids on the RCSwhey diet. 

The activities of the enzymes Δ^9^desaturase (D9C16) (*p* = 0.027) were higher in the ground corn and RCSmucilage treatments, and the activities of Δ^9^desaturase (D9C18) were higher (*p* = 0.049) in the RCSmucilage and RCSwhey treatments, while elongase was higher (*p* = 0.045) in the GC and RCSwater treatments.

## 4. Discussion

### 4.1. Growth and Physicochemical Composition of Goat Meat 

Dry matter (DM) intake was higher for goat kids fed with RCSmucilage (942 g/d) (Table 3). These results can be explained by the influence of the quality of the diet containing RCSmucilage (Table 1) that provided better utilization of the fiber and starch of the grain in the process of fermentation of the silage, since it contains the non-fiber carbohydrates (pectin) that are more digestible, increasing the rate of digestion of the diet ingredients due to its high degradability [34], and it consequently increased the NDF intake (350 g/d). The RCSmucilage diet promotes high energy content, without cause restriction in the intake of DM and other nutrients [35].

The replacement of GC by RCSmucilage in the diet of goat kids promoted a greater final body weight (27.6 kg). This can be justified by the increased palatability of the diet, mainly by the presence of cactus pear mucilage, which makes a diet different from the others because, with the formation of this compound, there is reduction in the losses through effluents of silage (Table 1) [36].

The pH of the meat of the goat kids feeding with RCSwater was above the values considered acceptable, which are between 5.4 and 5.8 for goat meat [37]. The decrease in pH occurs due to the conversion of muscle glycogen to lactic acid postmortem [38], which may suggest that these animals had less muscle glycogen reserve for the postmortem pH decline.

The similarity between energy sources and nutrients content of the different additives did not alter the meat color indexes, and this may be related to the proximity between the chemical composition of the diets (Table 1). In addition, the lack of effect of the diets on color indexes (L*, a*, b*, and chroma), WHC, and SF of the LL muscle could be related to the pH of the meat, which was also unaffected [39,40]. The meat color of the goat kids in this study showed the coordinates L*, a*, and b* within the ranges reported in the literature for male goat meat, which are around 28.8 to 49.74 for L*, 8.8 to 20.37 for a*, and 4.94 to 12.6 for b* [41,42,43,44]. This result indicates that the meat color of goat kids confined and fed dry corn or silage did not cause deleterious effects on meat color, which is associated with the quality of the final product and is extremely important at the moment of purchase by the consumer. The authors of Reference [45] also observed no differences in the L*, a*, and b* parameters in the meat of lambs fed dry or ensiled corn with high moisture.

Although goat meat is usually characterized as tough [46], in the present study, the shear force was considered low, and this result may have been influenced by the energy-rich diet, which provides the meat with a tendency to contain more soluble collagen and consequently makes the meat more tender; or it could also be due to the similar intramuscular fat content of the goat kids [47,48].

The cooking weight loss was greater for the meat from animals fed with GC and RCSmucilage (28.0 and 30.6 g/100 g, respectively) [49]. According to the literature [49,50,51,52], moisture and protein in goat meat were constant (approximately 75% moisture and 19 to 25% protein), as was also observed in our study (Table 4). Diets with high energy contents, when fed to confined animals, are also responsible for higher levels of fat deposition [53,54]. The lack of effect of diets on LL muscle fat content is associated with the dietary basis of the rations that was GC, and the influence of mucilage on the quality of starch GC was not enough to interfere with the deposition of muscle fat [55].

The ash content was higher (1.05/100 g) in the meat of animals fed with RCSwater and RCSmucilage. The values obtained in the present study for the centesimal composition of the meat were similar to those found in the literature for lamb meat [56,57], showing variations from 73.35 to 75.73%, 19.64 to 23.78%, 0.97 to 1.23%, and 2.14 to 5.3% for moisture, protein, ash, and lipids, respectively, characterizing meat of good nutritional quality.

### 4.2. Fatty Acid (FA) Profile of Goat Meat 

The FA profile of the intramuscular fat in the LL muscle of goat kids (Table 5) was affected by the different diets containing corn-grain silage with moisture additives and, in general, reflecting the FA profile of the diet [58]. Therefore, the RCSwhey diet provided lower concentrations of myristic acid (1.31%) in the muscle of the animals, and SFA is one of those responsible for promoting the accumulation of low-density lipoprotein, which, in turn, is a risk factor for the onset of cardiovascular diseases in humans since, in high amounts, it is undesirable, as it is related to the increase in total serum cholesterol and LDL [59,60].

In the present study, the most representative FA in the goat kids’ muscle was oleic acid, followed by stearic and palmitic fatty acids. However, palmitic content is generally higher than stearic in the LL muscle of goat kids [61]. This reduction is a beneficial response because 16:0 is a primary hypercholesterolemic FA in the human diet [62], while C18:0 has been shown to have a neutral effect on serum cholesterol in humans [63].

Higher concentrations of oleic acid were observed in the intramuscular fat of animals fed with RCSmucilage and RCSwater (41.1 and 41.2%). The oleic acid from its *cis* form is hypocholesterolemic and represents approximately 88% of the MUFA, acting in the reduction of the LDL and in the maintenance of HDL, and is related to the prevention of heart diseases [61,64]. The higher oleic acid values can be attributed to higher animal biosynthesis from stearic acid [65], explained by the higher activity of the C18:0 Δ^9^dessaturase enzyme in the LL muscle of these animals [66,67].

Usually, diets with a higher content of starch carbohydrates, such as the diet with RCSmucilage (Table 1), influence the proportions of the different volatile fatty acids produced in the rumen because they remain in the rumen for less time, leading to a shorter extent of biohydrogenation of the PUFA in the diet [68,69]. Another effect of diets containing a higher content of rapidly degrading starch in the rumen environment is the lowering of pH, which can result in incomplete biohydrogenation of linoleic and linolenic fatty acids [70]. This effect may increase the proportion of transient intermediates of PUFA biohydrogenation in these animals [71]. However, FA intermediates of biohydrogenation were not identified in the LL muscle of goat kids in the present research. 

Goat kids fed with RCSmucilage and RCSwhey showed an FA profile beneficial to health due to the higher concentrations of oleic acid, the presence of essential FA, and low concentrations of myristic acid when compared to the other diets, showing a change in these FAs [72]. Consequently, the meat of these animals had the highest levels of DFAs, which are important in a healthy diet. One of the main objectives of manipulating the FA profile of ruminant meat is to increase the content of PUFA, mainly those of the n-3 family, (Table 5) which have great importance in the prevention in most cardiovascular [73] and neurodegenerative diseases [74]. 

The concentrations of α-linolenic acid (ALA) and EPA in the LL muscle of the animals were increased with the RCSmucilage and RCSwhey diets (Table 5). ALA is the precursor of long-chain ω3 fatty acids, while linoleic acid is the precursor of long-chain ω6 fatty acids. Dietary ALA increases the accumulation of ALA and EPA in several depots, but the enrichment in DHA is minimal [62]. The RCSmucilage diet possibly had a higher linolenic acid content due to the cactus pear mucilage additive that contains in part of its cellular structures, especially the concentration in the EE of the diet [75], and this must have increased the proportions of this FA in the LL muscle of the animals as a result of less biohydrogenation in the rumen, thus allowing for the greater escape of these acids and, consequently, greater absorption of UFA in the small intestine.

This increase in ω3 family PUFA in the LL muscle of the goat kids provided by the RCSmucilage and RCSwhey diets reduced the ω6:ω3 ratio, which is an effect of the diet, as they play an important role in FA deposition, showing to be a good alternative to improve the production standard and enable value addition in the meat (Table 5); however, all treatments remained within the recommended value proposed by the Food and Agriculture Organization (FAO) [76] for human health, which is below 4, to reduce coronary complications, mainly blood clots, that can lead to heart diseases.

In the present study, the PUFA:SFA ratio found (0.23–0.25) was lower than that recommendation for a healthy diet, i.e., greater than 0.4 [77,78]. However, studies show that this ratio in meat is generally low, close to 0.1 [79], due to the biohydrogenation of unsaturated fatty acids in the diet by rumen microorganisms [61]. 

Regarding the low AI and TI values, more anti-atherogenic and anti-thrombogenic fatty acids are present in the lipids, and they may contribute to the prevention of cardiovascular diseases [80]. In the present study, although the GC, RCSwater, and RCSmucilage diets increased these indexes, these averages are considered low for meat, which are AI and TI less than 0.5 and 1.0, respectively [81]. The authors of Reference [29] consider AI and TI to be indicators of risk for cardiovascular disease for the consumer.

The presence of the ∆^9^dessaturase enzyme was higher because this enzyme tends to be more active in younger animals [82,83]. As for the activity of elongase, it can be explained by the lower amount of SFA, especially palmitic, thus showing a greater biosynthesis of FA, mainly oleic [82,84].

## 5. Conclusions

The diet that contained corn silage rehydrated with cactus pear mucilage improved the quality of the goat kids’ meat, increased the weight gain of the animals, and had a better ratio of fatty acids that are beneficial to human health, such as the ω3-family PUFA, thus making it possible to totally replace corn grain. The diet that contained corn silage rehydrated with whey showed better results regarding the nutraceutical compounds of the meat, showing that it is a nutritional alternative for the goat kids.

## Figures and Tables

**Table 1 animals-13-00031-t001:** Ingredient proportions and chemical composition of experimental diets and silages’ characteristics.

Variables	Experimental Treatments ^#^			
	Ground Corn	RCSwater	RCSmucilage	RCSwhey
Ingredients (g/kg DM)				
Tifton-85 hay	400	400	400	400
Soybean meal	180	170	170	170
Ground corn	400	0.00	0.00	0.00
Rehydrated corn-grain silage	0.00	415	415	415
Urea ^‡^	5.00	0.00	0.00	0.00
Mixture ^†^	15.0	15.0	15.0	15.0
Chemical composition (g/kg DM)				
Dry matter	920	819	828	814
Crude ash	73.8	64.5	66.3	64.8
Crude protein	158	150	151	151
Ether extract	28.6	26.9	31.7	26.3
Neutral detergent fiber_ap_ ^§^	430	379	398	375
Acid detergent lignin	44.7	33.0	33.2	33.0
Non-fiber carbohydrates	35.0	40.5	37.9	40.9
Metabolizable energy ^#^ (Mcal/kg)	2.60	2.36	2.38	2.38
Silage characteristics				
Dry matter	909	641	668	656
Crude protein	87.5	105	107	107
Crude ash	23.3	14.2	18.4	14.8
Neutral detergent fiber_ap_ ^§^	202	82.1	126	71.2
Acid detergent lignin	53.5	3.52	4.03	3.41
Dry matter recovery (g/100 g)	-	98.6	98.4	98.1
Gas losses (GL, g/100 g)	-	3.93	4.53	2.47
Effluent losses (EL, kg/ton)	-	3.52	3.31	4.16
pH	-	4.16	4.19	4.26
Ammonia nitrogen (N-NH_3_)	-	0.61	0.46	0.70

^#^ Rehydrated corn-grain silage with different moisture additives: water (RCSwater), cactus pear cladode mucilage (RCSmucilage), and whey (RCSwhey). ^‡^ Mixture of urea and ammonium sulfate at a ratio of 9:1. ^†^ Guaranteed levels (for active elements): 120 g calcium, 87 g phosphorus, 147 g sodium, 18 g sulfur, 590 mg copper, 40 mg cobalt, 20 mg chromium, 1800 mg iron, 80 mg iodine, 1300 mg manganese, 15 mg of selenium, 3800 mg of zinc, 300 mg of molybdenum, and a maximum of 870 mg of fluoride. ^§^ Corrected for ash and protein; ^#^ Calculated from the methodology [14].

**Table 2 animals-13-00031-t002:** Fatty acid profile of rehydrated corn-grain silages used in the goat kids feeding.

Fatty Acids (g/100 g FAMEs)	Experimental Treatments ^#^			
	Ground Corn	RCSwater	RCSmucilage	RCSwhey
Saturated (SFA)				
12:0	2.06	2.02	1.98	1.94
14:0	1.25	0.76	0.79	0.89
15:0	2.41	1.97	1.63	1.59
16:0	15.7	12.4	12.8	12.5
17:0	1.78	1.54	1.68	1.76
18:0	2.88	3.53	3.38	3.59
Monounsaturated (MUFA)				
14:1	10.0	10.1	9.77	9.76
15:1	4.85	4.66	3.85	3.89
16:1	2.71	2.33	2.34	2.38
17:1	2.49	2.40	2.38	2.44
18:1n9c	9.15	12.6	12.9	12.6
Polyunsaturated (PUFA)				
18:2n6t	0.46	0.46	0.46	0.46
18:2n6c	20.5	29.0	28.8	28.8
18:3n3	4.68	2.38	2.37	2.38
18:3n6	4.36	4.17	4.53	4.18
Other ^1^	14.7	9.69	10.4	10.9
SFA	36.7	29.6	30.2	30.6
MUFA	29.2	32.1	31.2	31.1
PUFA	34.1	38.3	38.6	38.3

^#^ Rehydrated corn-grain silage with different moisture additives: water (RCSwater), cactus pear cladode mucilage (RCSmucilage), and whey (RCSwhey). ^1^ Sum of minor FAs: 20:2, 22:0, 22:1, 23:0, and 24:0.

**Table 3 animals-13-00031-t003:** Performance of goat kids fed with dry corn grain and ground-corn-grain silage rehydrated with different additives.

Item	Experimental Treatments ^#^				SEM ^1^	*p*-Value ^2^
	Ground Corn	RCSwater	RCSmucilage	RCSwhey		
Initial body weight (kg)	16.4	16.4	16.4	16.4	-	-
Final body weight (kg)	26.4 ^a,b^	24.7 ^b^	27.6 ^a^	25.3 ^b^	1.54	0.033
Nutrient intake (g/d)						
Dry matter	882 ^a,b^	842 ^b^	934 ^a^	828 ^b^	39.0	0.035
Crude protein	146	139	150	138	2.91	0.22
Neutral detergent fiber	350 ^a^	288 ^b^	340 ^a^	290 ^b^	8.61	0.0009
Metabolizable energy ^3^ (Mcal/kg)	2.29	1.99	2.14	1.94	0.065	0.086

^#^ Rehydrated corn-grain silage with different moisture additives: water (RCSwater), cactus pear cladode mucilage (RCSmucilage), and whey (RCSwhey). ^1^ Standard error of the mean; ^2^
*p*-Value significant at *p* < 0.05. Means followed by the same letter in the row do not differ statistically according to the Tukey’s test at 5% probability. ^3^ Calculated from the methodology Weiss (1993).

**Table 4 animals-13-00031-t004:** Chemical composition and physical properties of the longissimus lumborum muscle of goat kids fed with dry corn grain and ground-corn-grain silage rehydrated with the different additives.

Item	Experimental Treatments ^#^				SEM ^1^	*p*-Value ^2^
	Ground Corn	RCSwater	RCSmucilage	RCSwhey		
Final pH (24 h)	5.61	5.95	5.54	5.48	0.060	0.072
Color indexes						
Lightness (L*)	39.3	39.3	39.1	40.4	0.34	0.49
Red (a*)	16.2	15.9	16.4	16.0	0.13	0.59
Yellow (b*)	5.93	6.06	5.52	6.02	0.14	0.52
Croma (C*)	17.3	17.1	17.4	16.9	0.12	0.55
WRC ^3^ (g/100 g)	79.4	77.7	76.2	77.2	0.57	0.35
CWL ^4^ (g/100 g)	28.0 ^a^	20.8 ^b^	30.6 ^a^	22.2 ^b^	1.03	0.0001
SF ^5^ (N)	6.72	6.35	6.83	5.16	0.36	0.39
Chemical composition (g/100 g)						
Moisture	74.6	73.9	74.5	74.6	0.21	0.63
Protein	20.8	21.8	22.1	21.7	0.19	0.25
Lipids	2.58	2.63	2.51	2.69	0.16	0.98
Ash	0.99 ^b^	1.05 ^a^	1.05 ^a^	0.99 ^b^	0.011	0.027

^#^ Rehydrated corn-grain silage with different moisture additives: water (RCSwater), cactus pear cladode mucilage (RCSmucilage), and whey (RCSwhey). ^1^ Standard error of the mean; ^2^
*p*-value significant at *p* < 0.05. Means followed by the same letter in the row do not differ statistically according to the Tukey’s test at 5% probability. ^3^ Water-holding capacity. ^4^ Cooking weight loss. ^5^ Shear force Warner–Bratzler (Newton’s force).

**Table 5 animals-13-00031-t005:** Fatty acid profile of longissimus lumborum muscle of goat kids fed with dry corn grain and ground-corn-grain silage rehydrated with the different additives.

Fatty Acids (% Total FAMEs)	Experimental Treatments ^#^				SEM ^1^	*p*-Value ^2^
	Ground Corn	RCSwater	RCSmucilage	RCSwhey		
Saturated fatty acids (SFAs)						
C4:0	1.95 ^a^	1.85 ^a^	1.46 ^b^	1.06 ^c^	0.064	0.018
C6:0	1.71 ^a^	1.09 ^b^	1.98 ^a^	1.04 ^b^	0.73	0.045
C8:0	1.61 ^a^	1.44 ^a^	1.01 ^b^	1.52 ^a^	0.43	0.041
C10:0	1.26	1.66	1.58	1.66	0.44	0.32
C14:0	1.86 ^a^	2.18 ^a^	1.73 ^a^	1.31 ^b^	0.26	0.014
C15:0	2.03 ^a^	1.17 ^c^	1.55 ^b^	1.57 ^b^	0.86	0.029
C16:0	14.2	14.1	15.1	15.4	1.94	0.96
C17:0	1.82	1.96	1.36	1.42	0.33	0.46
C18:0	16.5	16.9	16.7	16.4	1.75	0.65
Other SFAs	1.53	1.46	1.53	1.49	0.32	0.28
Monounsaturated fatty acids (MUFAs)						
C16:1	1.54	1.49	1.64	1.59	0.23	0.43
C17:1	0.75	0.86	0.78	0.79	0.075	0.95
C18:1 ω9	39.3 ^b^	39.0 ^b^	41.1 ^a^	41.2 ^a^	3.72	0.046
C18:1 c-11 cis	1.12	1.14	1.24	1.30	0.05	0.081
C22:1 ω9	2.01 ^b^	2.07 ^b^	1.85 ^c^	3.30 ^a^	0.31	0.0001
Other MUFAs	0.65	0.69	0.59	0.61	0.042	0.53
Polyunsaturated fatty acids (PUFAs)						
C18:2 ω6	3.63	3.49	3.37	3.72	0.36	0.95
C18:2 t-9,12	0.35	0.33	0.35	0.34	0.023	0.74
C18:3 ω3	0.44 ^b^	0.42 ^b^	0.62 ^a^	0.43 ^b^	0.03	0.019
C20:2 ω6	0.45 ^b^	0.43 ^b^	0.49 ^a,b^	0.62 ^a^	0.04	0.033
C20:3 ω3	0.41 ^b^	0.44 ^b^	0.29 ^c^	0.73 ^a^	0.07	<0.0001
C20:4 ω6	1.34	1.31	1.33	1.32	0.934	0.46
C20:5ω3 (EPA) ^4^	0.44 ^b^	0.43 ^b^	0.58 ^a^	0.57 ^a^	0.033	0.010
C22:6ω3 (DHA) ^4^	0.39	0.38	0.36	0.41	0.029	0.47
Other PUFAs	2.75 ^b^	3.71 ^a^	2.96 ^a,b^	2.25 ^c^	0.118	<0.0001
Sum of groups						
∑SFA ^4^	44.5 ^a^	43.8 ^a,b^	44.0 ^a,b^	42.9 ^b^	2.83	0.012
∑MUFA ^4^	45.4 ^b^	45.3 ^b^	47.2 ^a,b^	48.8 ^a^	3.73	<0.0001
∑PUFA ^4^	10.2 ^b^	10.9 ^a^	10.4 ^b^	10.3 ^b^	0.41	0.001
ω6 ^4^	5.42	5.23	5.19	5.66	0.36	0.92
ω3 ^4^	1.68 ^b^	1.67 ^b^	1.85 ^a,b^	2.14 ^a^	0.09	<0.0001
Ratios						
MUFA:SFA	1.02 ^b^	1.03 ^b^	1.07 ^a,b^	1.14 ^a^	0.113	0.017
PUFA:SFA	0.23 ^b^	0.25 ^a^	0.24 ^a,b^	0.24 ^a,b^	0.012	<0.0001
ω6:ω3	3.23 ^a^	3.13 ^a^	2.81 ^b^	2.64 ^b^	0.194	0.006
Nutraceutical Compounds						
Desirable fatty acids	72.1 ^b^	73.1 ^b^	74.3 ^a^	75.6 ^a^	4.31	0.044
h:H ^3^ ratio index	2.67	2.61	2.69	2.69	0.10	0.48
Atherogenicity index	0.41 ^a^	0.44 ^a^	0.41 ^a^	0.36 ^b^	0.03	0.044
Thrombogenicity index	0.62 ^a^	0.63 ^a^	0.62 ^a^	0.58 ^b^	0.06	<0.0001
Enzymatic activity						
Δ^9^desaturase (D9C16) ^5^	9.78 ^a^	9.56 ^b^	9.80 ^a^	9.36 ^c^	1.97	0.027
Δ^9^desaturase (D9C18) ^5^	70.4 ^b^	69.8 ^b^	71.1 ^a^	71.5 ^a^	1.75	0.0497
Elongase ^5^	78.0 ^a^	78.2 ^a^	77.5 ^b^	77.2 ^b^	1.02	0.045

^#^ Rehydrated corn-grain silage with different moisture additives: water (RCSwater), cactus pear cladode mucilage (RCSmucilage), and whey (RCSwhey). ^1^ SEM = standard error of the mean; ^2^
*p*-value significant at *p* < 0.05. Means followed by the same letter in the row do not differ statistically according to the Tukey’s test at 5% probability. ^3^ Hypocholesterolemic:hypercholesterolemic fatty acids ratio. ^4^ EPA = eicosapentaenoic acid; DHA = docosahexaenoic acid; Total SFAs, sum of saturated fatty acids; Total MUFAs, sum of monounsaturated fatty acids; Total PUFAs, sum of polyunsaturated fatty acids; Total ω3, sum of ω3 fatty acids, including EPA and DHA; Total ω6, sum of ω6 fatty acids. ^5^ Enzymatic activities: D9C16 = [C16:1/(C16:0 + C16:1)] × 100, D9C18 = [(C18:1 cis–9)/(C18:0 + C18:1 cis–9)] × 100 and elongase = [(C18:0+ C18:1 cis–9)/(C16:0 + C16:1 + C18:0 + C18:1 cis–9)] × 100.

## Data Availability

Not applicable.

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
