# Peer review of "Physicochemical Composition and Fatty Acid Profile of Goat Kids’ Meat Fed Ground-Corn-Grain Silage Rehydrated with Different Additives"

_animals, 2022, doi:10.3390/ani13010031_

Round 1

Reviewer 1 Report

The manuscript entitled "Physicochemical composition and fatty acid profile of goat kids meat fed ground corn grain silage rehydrated with different additives" by Luciana Viana Diogénes et al (animals-2065466) has been assessed.  Authors evaluated the effects of different additives for ground corn grain silage replacing dry ground corn grain on growth, physicochemical composition and fatty acid profile of goat kids meat. The experimental design is well and the language is good. I have some concerns on the current presentation, show as follows:

(1)  Simple Summary: Line 21-23, water (RCSwater), cactus pear cladode mucilage (RCSmucilage) and whey (RCSwhey); and replace ground corn (GC) by rehydrated corn grain silage in the finishing of goat kid’s diet. Here the abbreviation of treatments is too long, please find another one to replace.

(2) Abstract: Line 30-31, thirty-two crossbred goat kids 30 (16 ± 2.50 kg initial body weight) were assigned to experimental treatments. Here, the average initial body weight is 16 kg, however, the initial body weight showed in Table 3 were all greater than 16.4 kg. Please check the data.

(3) Abstract: the whole description was too long and needs improvement to highlight the main points: backgroud, aim, method, results, and conclusion.

(4) Introduction: Some related works are missed in this section, please find them and described them in a acceptable way.

(5) Result: Table 5, Enzymatic activity, the detailed calculations should be also provided in table note. Please find similar in other places.

(6) Discussion: the current presentation of this section seems a bit chaotic, please divide them into 4.1, 4.2, ...to highlight issues.

Author Response

Dear Animals Editorial Office

We have thanked the adjustments suggested and we really appreciate the attention of the reviewer’s contribution in the analysis and correction of this paper. We have modified the manuscript according to the review points, and we are forwarding the updated version of our paper.  We are of course always available, and we are pleased to provide any clarification that may be required. All corrections were addressed, as you can see below and in the attached file. Answers to the questions are provided below, and all the changes in the manuscript have been red highlighted.  

Sincerely yours,

Leilson Rocha Bezerra

Reviewer #1

Comments for the author

Correction and answers

The manuscript entitled "Physicochemical composition and fatty acid profile of goat kids meat fed ground corn grain silage rehydrated with different additives" by Luciana Viana Diogénes et al (animals-2065466) has been assessed.  Authors evaluated the effects of different additives for ground corn grain silage replacing dry ground corn grain on growth, physicochemical composition and fatty acid profile of goat kids meat. The experimental design is well and the language is good. I have some concerns on the current presentation, show as follows:

Thank you. We have thanked the adjustments suggested and we really appreciate your attention in the analysis and correction of this paper. We have modified the manuscript according to the review points, and we are forwarding the updated version of our paper.  We are available, and we are pleased to provide any clarification that may be required. All corrections were addressed, as you can see below and in the attached file.

(1)  Simple Summary: Line 21-23, water (RCSwater), cactus pear cladode mucilage (RCSmucilage) and whey (RCSwhey); and replace ground corn (GC) by rehydrated corn grain silage in the finishing of goat kid’s diet. Here the abbreviation of treatments is too long, please find another one to replace.

(1)  Line 21-23: Simple Summary: We have replaced the abbreviation of treatments here. they are unnecessary. please see if this is better.

(2) Abstract: Line 30-31, thirty-two crossbred goat kids 30 (16 ± 2.50 kg initial body weight) were assigned to experimental treatments. Here, the average initial body weight is 16 kg, however, the initial body weight showed in Table 3 were all greater than 16.4 kg. Please check the data.

(2) Line 31: Abstract: Please excuse me. This was corrected to 16.4 kg.

(3) Abstract: the whole description was too long and needs improvement to highlight the main points: backgroud, aim, method, results, and conclusion.

(3) Abstract: We have rewritten the Abstract section to highlight the main points: background, aim, method, results, and conclusion. We have added in the abstract a suggestion concerning the most suitable for improve meat quality too.

(4) Introduction: Some related works are missed in this section, please find them and described them in a acceptable way.

(4) Thanks for the explanation. We have rewritten to improve the explanation in the Introduction section and have added works more adherent to the subject according to suggestion.

(5) Result: Table 5, Enzymatic activity, the detailed calculations should be also provided in table note. Please find similar in other places.

(5) Lines 370-372, Result: Table 5, We have added equations detailed of enzymatic activity, the in table note. This was added too in M&M.

(6) Discussion: the current presentation of this section seems a bit chaotic, please divide them into 4.1, 4.2, ...to highlight issues.

(6) Please excuse us. We subdivided the Results and Discussion into topics. We appreciate the suggestion and have since rewritten various parts of the Discussion, removing unnecessary and sparsely explanatory text, and rewriting other parts to improve understanding. Please see if now the discussion is more explanatory and succinct.

Reviewer 2 Report

Dear Authors

The introduction should be improved, because your study brings relevant information to the scientific field?

The discussion should also be improved: new references should be added. The authors should justify the findings of the experiment. Authors should compare their studies with other recent studies, and the reasons for the differences.

Author Response

Manuscript ID: animals-2065466

Original title: Physicochemical composition and fatty acid profile of goat kids meat fed ground corn grain silage rehydrated with different additives

Dear Animals Editorial Office

We have thanked the adjustments suggested and we really appreciate the attention of the reviewer’s contribution in the analysis and correction of this paper. We have modified the manuscript according to the review points, and we are forwarding the updated version of our paper.  We are of course always available, and we are pleased to provide any clarification that may be required. All corrections were addressed, as you can see below and in the attached file. Answers to the questions are provided below, and all the changes in the manuscript have been red highlighted.  

Reviewer #2

Comments for the author

Correction and answers

The introduction should be improved, because your study brings relevant information to the scientific field?

Thanks for the explanation. We have rewritten to improve the explanation in the Introduction section and have added works more adherent to the subject according to suggestion.

The discussion should also be improved: new references should be added. The authors should justify the findings of the experiment. Authors should compare their studies with other recent studies, and the reasons for the differences.

Discussion

Please excuse us. We appreciate the suggestion and have since rewritten various parts of the Discussion, removing unnecessary and sparsely explanatory text, and rewriting other parts to improve understanding. Please see if now the discussion is more explanatory and succinct.

Reviewer 3 Report

In the paper the effects of the inclusion of ground corn grain silage treated with different additives has been evaluated on the meat characteristics. The paper it’s interesting and very well structured. I suggest only few revisions:

Abstract 

Please add in the abstract the suggestion concerning the most suitable for improve meat quality

Table 3

Have you analyzed the initial body weight of the animals? Why do you not report the p value and the SEM for this data?

Table 5

Please report all analyzed the fatty acids, also the minor constituents.

Lines 501-503: the differences were very poor 9.78 vs. 9.80%, please remove this sentence or remodulate it.

Discussion

Revise the discussion and discuss only significant results.

Author Response

Manuscript ID: animals-2065466

Original title: Physicochemical composition and fatty acid profile of goat kids meat fed ground corn grain silage rehydrated with different additives

Dear Animals Editorial Office

We have thanked the adjustments suggested and we really appreciate the attention of the reviewer’s contribution in the analysis and correction of this paper. We have modified the manuscript according to the review points, and we are forwarding the updated version of our paper.  We are of course always available, and we are pleased to provide any clarification that may be required. All corrections were addressed, as you can see below and in the attached file. Answers to the questions are provided below, and all the changes in the manuscript have been red highlighted.  

Sincerely yours,

Leilson Rocha Bezerra

Reviewer #3

Comments for the author

Correction and answers

In the paper the effects of the inclusion of ground corn grain silage treated with different additives has been evaluated on the meat characteristics. The paper it’s interesting and very well structured. I suggest only few revisions:

Thank you. We have thanked the adjustments suggested and we really appreciate your attention in the analysis and correction of this paper. We have modified the manuscript according to the minor points, and we are forwarding the updated version of our paper.  We are available, and we are pleased to provide any clarification that may be required. All corrections were addressed, as you can see below and in the attached file.

Abstract 

Please add in the abstract the suggestion concerning the most suitable for improve meat quality

Abstract 

We have rewritten the Abstract section to highlight the main points: background, aim, method, results, and conclusion. We have added in the abstract a suggestion concerning the most suitable for improve meat quality too.

Table 3

Have you analyzed the initial body weight of the animals? Why do you not report the p value and the SEM for this data?

Table 3

Thank you for your consideration. The variable initial body weight of the animals is a variable that we control to ensure homogeneity in the distribution of animals in the treatments. So it wouldn't make sense for us to analyze it because we distributed it in blocks so that there are no differences on average

Table 5

Please report all analyzed the fatty acids, also minor constituents.

Table 5

Please report all analyzed the fatty acids, also minor constituents.

Lines 501-503: the differences were very poor 9.78 vs. 9.80%, please remove this sentence or remodulate it.

Lines 501-503: This was removed.

Discussion

Revise the discussion and discuss only significant results.

Discussion

Please excuse us. We appreciate the suggestion and have since rewritten various parts of the Discussion, removing unnecessary and sparsely explanatory text, and rewriting other parts to improve understanding. Please see if now the discussion is more explanatory and succinct.

Round 2

Reviewer 1 Report

It is still strange to see "with an average weight of 16.4 ± 2.5 kg, whilst the average Initial body weight in Table 3 were all greater than 16.4, how to understand? I am still confused, please check!

Author Response

Dear Reviewer

We have thanked the adjustments suggested and we really appreciate the attention of the reviewer’s contribution in the analysis and correction of this paper.

You are right. The initial body weight data in Table 3 are wrong and it have been corrected. As we have reported, the initial weight of all groups after the formation of experimental blocks was 16.4 with a small standard deviation (SD). We have corrected the initial body weight values in Table 3.

Answers to the questions are provided below, and all the changes in the manuscript have been red highlighted.  

Sincerely yours,

Leilson Rocha Bezerra
